# An Undercurrent off the East Coast of Sri Lanka

Arachaporn Anutaliya[1], Uwe Send[1], Julie L. McClean[1], Janet Sprintall[1], Luc Rainville[2], Craig Lee[2], S. U. Priyantha Jinadasa[3], Alan J. Wallcraft[4], E. Joseph Metzger[4]

[1]Scripps Institution of Oceanography, La Jolla, California, USA
[2]Applied Physics Laboratory, University of Washington, Seattle, Washington, USA
[3]National Aquatic Resources Research and Development Agency, Colombo, Sri Lanka
[4]Naval Research Laboratory, Stennis Space Center, Mississippi, USA

*Correspondence to*: Arachaporn Anutaliya (aanutali@ucsd.edu)

**Abstract.** The existence of a seasonally varying undercurrent along 8° N off the east coast of Sri Lanka
is inferred from shipboard hydrography, Argo floats, glider measurements, and two Ocean General Circulation Model simulations. Together, they reveal an undercurrent below 100-200 m flowing in the opposite direction to the surface current that is most pronounced during boreal spring and summer and switches direction between these two seasons. The volume transport of the undercurrent (200-1000 m layer) can be more than 10 Sv in either direction, exceeding the transport of 1-6 Sv carried by the
surface current (0-200 m layer). The undercurrent transports relatively fresh water southward during spring, while it advects more saline water northward along the east coast of Sri Lanka during summer. Although the undercurrent is potentially a pathway of salt exchange between the Arabian Sea and the Bay of Bengal, the observations and the OGCMs suggest that the salinity contrast between seasons and between the boundary current and interior is less than 0.09 in the subsurface layer, suggesting a small
salt transport by the undercurrent of less than 4% of the salinity deficit in the Bay of Bengal.

## 1 Introduction

Knowledge of the circulation in the southern Bay of Bengal (BoB) is crucial to understanding the contrasting salinity distributions of the Arabian Sea (AS) and the BoB since it controls the amount of water and salt exchanged between the two ocean basins. Observations and modeling studies have
confirmed the role of the surface current along the Sri Lankan east coast in distributing mass and salt between the AS and the BoB. Drifters indicate that a strong surface-intensified current to the east of Sri Lanka during the northeast monsoon (November- January) transports low-salinity water out of the BoB toward the AS around the south coast of Sri Lanka (Wijesekera et al., 2015). An ocean general circulation model (OGCM) study also shows that significant mass and salt transport occur between the
AS and the BoB along the eastern and southern coast of Sri Lanka (Jensen et al., 2016).

The surface current to the east of Sri Lanka is influenced by local alongshore winds, remote winds in the vicinity of the northern and eastern boundaries of the BoB, equatorial waves, and interior Ekman pumping (Yu et al., 1991; Shetye et al., 1993; McCreary et al., 1996; Shankar et al., 1996;
Vinayachandran and Yamagata, 1997). This current has a strong seasonal pattern (Shankar et al., 2002; Lee et al., 2016) with recirculation loops that are highly variable in time and space (Durand et al.,

2009). Some of the recirculations appear seasonally, such as the Sri Lanka Dome (SLD), a cyclonic eddy that is well developed in July (Vinayachandran and Yamagata, 1997). The SLD is driven by Rossby waves radiating from the eastern boundary and intensified Ekman pumping inside the BoB (Vinayachandran et al., 1999; Shankar et al., 2002; de Vos et al., 2014). The SLD propagates westward (Wijesekera et al., 2016b) toward the east coast of Sri Lanka resulting in a southward coastal surface flow during early summer. In October, the prevailing wind in the BoB starts reversing direction and blows southwestward. This marks the start of the northeast monsoon when the local wind along the east coast of India drives the East India Coastal Current (EICC) southward with speeds exceeding 1 m s$^{-1}$ extending from the east coast of India southward to the southern tip of Sri Lanka (Shetye et al., 1996; Wijesekera et al., 2015).

The *subsurface* circulation in this region is also potentially important as high-salinity water from the AS can be subducted beneath the fresher surface water originating from river runoff and precipitation in the northern BoB (Rao & Sivakumar, 2003; Sengupta et al., 2006; Vinayachandran et al., 2013; Gordon et al., 2016; Jensen et al., 2016); this is evident in observations and model studies both during the northeast (Wijesekera et al., 2015) and southwest monsoons (June- August) (Wijesekera et al., 2016b). Little is known about the subsurface structure of the boundary current along the east coast of Sri Lanka. Mooring observations show a reversing subsurface current occurring off the southern coast of Sri Lanka; it is most distinct during boreal spring and summer (Schott et al., 1994). In addition, reversal of the EICC along the east coast of India is observed below 100 m that is southward during the southwest monsoon and northward during the northeast monsoon, with the winter undercurrent being a more permanent feature (Mukherjee et al., 2014). The direction of this undercurrent is in good agreement with findings from a linear, continuously stratified ocean model (LSM) (McCreary et al., 1996). The LSM also suggests the presence of an undercurrent along the Sri Lankan east coast (centered at 8$^{\circ}$N) that reverses its direction twice a year with model speeds ranging from 6 cm s$^{-1}$ equatorward during boreal spring to 8 cm s$^{-1}$ poleward during summer. This undercurrent has not been observed before and will be the focus of this study. We will investigate the vertical structure and seasonal variability of subsurface flows in the boundary current system off the eastern coast of Sri Lanka using OGCMs as well as observations from shipboard hydrography, Argo floats, and glider measurements. Knowledge of the vertical structure, variability, and associated dynamics of the boundary current will contribute to a better understanding of mass and salt exchanges in the northern Indian Ocean.

## 2 Data and Methodology

Shipboard hydrography and Argo Conductivity-Temperature-Depth (CTD) profiles, together with satellite altimeter-derived surface absolute dynamic topography (ADT), were used to determine the vertical structure of the current east of Sri Lanka. There were 13 shipboard hydrography and 116 Argo profiles available between 7-9° N and within 130 km of the Sri Lankan east coast that sampled to at least 1000 m depth (Figure 1a). The hydrographic profiles were collected over the period of 2008-2017 while the Argo profiles were collected over 2007-2017. The gridded delayed-time ADT products were derived from Archiving, Validation, and Interpretation of Satellite Oceanographic (AVISO) Data

(Ducet et al., 2000) which are available over the period of 1993-2016 with spatial and temporal resolutions of ¼° latitude × ¼° longitude and 7 days, respectively.

The hydrography-Argo based mean seasonal absolute geostrophic meridional velocity profiles were calculated by combining sheared velocity profiles from hydrography and Argo measurements and surface velocity from the ADT products. The number of hydrography and Argo profiles available were too few to calculate robust estimates of monthly meridional geostrophic velocity. Hence, to derive the seasonal sheared velocity profiles, we used 2-month (*i.e.* bimonthly) sliding mean dynamic heights calculated from hydrography-Argo profiles located to the west and east of the central position of the boundary current (Figure 1a); the position of the boundary current was determined from a longitude-depth velocity transect along 8° N derived from glider measurements and OGCMs (for example, Figure 2). The low-pass filtered geostrophic meridional sheared velocity profile is thus proportional to the corresponding difference between the eastern and western filtered dynamic heights. The number of hydrography and Argo profiles available in each bimonth period varied from 3 to 14 in the western region and from 4 to 24 in the eastern region (Figure 1b). The equally low-passed surface meridional velocity from the satellite ADT along 8° N between 81.75° E-82.5° E was added to the sheared profiles to yield absolute geostrophic meridional velocity profiles.

Gliders were deployed to the east of Sri Lanka between 7° N and 9° N to collect salinity and temperature profiles from the surface to 1000 m depth during February 2014–January 2016 (Lee et al., 2016) (Figure 1a). The geostrophic velocities from the glider measurements are referenced in two ways. Glider temperature and salinity profiles are used to calculate geostrophic velocity across 8° N referenced to the glider depth-averaged velocity to examine the vertical structure of the undercurrent (e.g. Figure 2). Note that profiles sampled by the gliders were sporadic in time; the gliders took approximately 2-7 days to complete the transect and then could take 5-30 days until the same transect was repeated. Thus, the glider velocity referenced to the depth-averaged current is less suitable to estimate the seasonal evolution of the circulation along Sri Lankan east coast (*i.e.* Figure 3). Instead, temperature and salinity profiles were used in the same way as the hydrography and Argo profiles to calculate bimonthly geostrophic meridional velocity.

Two global strongly eddy active OGCMs were used to study the vertical structure and seasonal variation of the boundary current, as well as its associated dynamics: 0.1° Parallel Ocean Program (POP) and 0.08° HYbrid Coordinate Ocean Model (HYCOM). Their horizontal resolutions correspond to 11 km and 9 km at 8°N, respectively. POP is a three-dimensional, $z$-level, primitive equation model (Smith et al., 2010). It was configured on a global tripole grid with 42 vertical levels and partial bottom cells. Its vertical grid spacing near the surface (top 50m) is roughly 10 m. The POP simulation was initialized from year 30 of an earlier POP run that used the same set-up but was driven by a monthly climatology of Co-ordinated Ocean Reference Experiments (CORE) atmospheric surface fluxes (Maltrud et al., 2010). The subsequent POP simulation was forced by interannually-varying CORE-II fluxes for 1990-2007 (Delman et al., 2015). The POP output analyzed here consists of three-dimensional daily-averaged velocities and salinity for 1995-2007. HYCOM has a hybrid vertical coordinate with isopycnal coordinates in the open stratified ocean, a terrain-following coordinate for

coastal regions, and a *z*-level coordinate in the mixed layer (Metzger et al., 2010). It was configured on a global tripole grid with 41 vertical layers. Vertical grid spacing very close to the surface is roughly 1 m increasing to 8 m over 36-85 m. The simulation was forced by 3-hourly Navy Operational Global Atmospheric Prediction System (NOGAPS) fields (Rosmond et al., 2002). However, the archived velocity fields used here are monthly and cover a shorter period compared to POP of 2003-2012.

## 3 The Circulation east of Sri Lanka

### 3.1 Vertical structure

A zonal transect along 8° N is used to examine the circulation off the eastern coast of Sri Lanka. Both observations and OGCMs show a seasonally reversing surface current confined to the upper 100-200 m of the water column (Figures 2, 3). A subsurface current that flows in the opposite direction to that of the surface water, defined here as an undercurrent, is observed from glider measurements and the OGCMs particularly in March and June (Figure 2). As noted above, geostrophic velocities derived from glider measurements, during March 18-30, 2014 (Figure 2a) and May 28–June 8, 2014 (Figure 2d), are referenced to the depth-averaged velocity measured directly by the gliders. Velocity transects from HYCOM (Figure 2b, e) and POP (Figure 2c, f) were constructed from monthly averages of total velocity in March and June for the periods of 2003-2012 and 1995-2007, respectively.

The model transects show reasonable agreement with that from the gliders in regard to the vertical structure of both the surface current and undercurrent. In March, the core of the observed undercurrent is located about 55 km from the coast and has a maximum speed of 16-25 cm s$^{-1}$ southward (Table 1). HYCOM and POP have maximum flows in the reverse direction to the surface *i.e.* southward, at 570 and 268 m, respectively (Figure 2a-b) while glider measurements show maximum subsurface flow at the deepest measured depth of 1000 m (Figure 2c). In June, a northward undercurrent is observed below approximately 250 m with a maximum speed greater than 20 cm s$^{-1}$ at about 900 m (Figure 2d-f; Table 2). The summer undercurrent is approximately 165 km wide in the OGCMs. The velocity section derived from gliders shows a wider undercurrent of 210 km with two cores: a deep core that is approximately 100 km wide centered at 82.2° E and a shallow core at 400 m near 82.75° E (Figure 2d). This double-core feature is also evident in some individual model years, but is not present in the multi-year mean (Figure 2e, f).

Investigation of individual years of HYCOM and POP velocity fields reveals that although the location of the undercurrent core varies in depth and longitude, it is nearly always confined to the west of 82.5° E. Thus, this longitude limit will be used in subsequent averages over the undercurrent regime, and is indicated by the dashed line in Figure 2. The spring undercurrent exists in all years in the model simulations except for in 2007 and 2011 in HYCOM when an El Niño occurs in the preceding year. POP also shows a weakening of the spring undercurrent following an El Niño event, except in 1998 when the northward-flowing surface current is absent and the southward current extends from the surface to approximately 550 m depth.

### 3.2 Seasonal variation

Bimonthly mean meridional velocities computed from hydrography and Argo profiles and glider measurements are shown in Figure 3a and 3b respectively. These observed geostrophic velocities are all referenced to surface velocity from the ADT. The model monthly mean total meridional velocities (Figure 3c, d) were constructed from the OGCMs by averaging the monthly mean velocity across 8° N along the transect from the eastern coast of Sri Lanka to 82.5° E (depicted by the magenta transect in Figure 1), which captures most of the undercurrent signal (Figure 2; Table 1, 2).

### 3.2.1 Surface circulation

Both surface and subsurface currents have strong seasonal variability (Figure 3). The surface current reverses its direction twice a year in agreement with previous studies (Yu et al., 1991; McCreary et al., 1996; Shetye et al., 1996; Vinayachandran and Yamagata, 1997; Vinayachandran et al., 1999; Eigenheer and Quadfasel, 2000; Durand et al., 2009; de Vos et al., 2014; Wijesekera et al., 2015; Lee et al., 2016). The mean meridional surface current is northward during boreal spring (Figure 2a-c, Figure 3) as the westward flowing North Equatorial Current often bifurcates at the east coast of Sri Lanka at approximately 7.5° N splitting into a northward and southwestward branch (Shetye et al., 1993; Hacker et al., 1998). The SLD emerges during the southwest monsoon. Its western branch is responsible for the observed southward flow along the eastern coast of Sri Lanka in that season (Figure 2d-f, Figure 3). During boreal fall, the surface current is northward. Similarly, northward surface flow was also observed in mooring measurements in the southern BoB in late July of 2015 at 85.5° E (Wijesekera et al., 2016b). During the winter, the southward-flowing EICC extends from the Indian east coast to the southern tip of Sri Lanka (Schott et al., 1994; Wijesekera et al, 2015) resulting in strong southward flow at 8° N (Figure 3).

### 3.2.2 Subsurface circulation

An undercurrent flowing in the opposing direction to the surface current is a prominent and consistent feature among the observations and OGCMs in boreal spring (beginning of February to mid-April) and summer (beginning of June to mid-August) (Figure 3). In boreal spring, the southward-flowing undercurrent extends from approximately 200 to 900 m depth with a velocity maximum core in the range of 300-600 m depth. The velocities derived from hydrography and Argo profiles and glider measurements confirm the existence of the undercurrent below 150-250 m, although the magnitudes are not consistent (Figure 3). This may be due to interannual variation in intensity and position of the subsurface reversed flow, combined with the sporadic sampling of the hydrography and Argo data.

During the southwest monsoon, the northward-flowing undercurrent is apparent in both observations and the OGCMs (Figure 3). The undercurrent starts developing in May and reaches its maximum speed in June. The OGCMs also suggest an upward phase propagation during summer and early fall (May-October) with a speed of approximately 1.5 m day$^{-1}$ in the upper 200 m of the water column (Figure 3c,

d). This feature is also evident in the geostrophic velocity derived from hydrography-Argo profiles in the upper 100 m of the water column (Figure 3a).

To gain a better understanding of the subsurface circulation, we examine the POP velocity fields at 729 m (Movie S1), which encompasses the depth location of the undercurrent core in boreal spring and summer (Figures 2, 3; Tables 1, 2). In March, the undercurrent is strong and confined to 82.5° E along the entire Sri Lankan east coast (Movie S1). The undercurrent usually turns eastward around 6° N to combine with an eastward-flowing current along the Sri Lankan south coast, as shown in the simplified schematic derived from the POP 729 m velocity fields (Figure 4a). However, in some years (e.g. 1995, 1999, 2005, and 2007), the undercurrent turns westward around the southern coast of Sri Lanka and forms a narrow current, about 50-70 km wide, on the northern side of the eastward flow (Movie S1; Figure 4a). Note that a similar circulation is observed in the POP velocity fields at 318 m in March, albeit the flow is stronger. In boreal summer, there is a convergence of the eastward-flowing current along the south coast of Sri Lanka and a westward-flowing current in the southwestern BoB at approximately 7° N, 82.25° E (Movie S1), which is also shown in the simplified schematic (Figure 4b). Both OGCMs agree that the convergence produces the undercurrent flowing north along the east coast of Sri Lanka that injects relatively saline water into the BoB. In addition, an anticyclonic eddy often develops along the east coast, trapping local saline water inside, which is then propagated northward with the eddy (Movie S1).

During boreal fall (mid-August to end of October) and winter (beginning of November to end of January), flows in the subsurface layer are less consistent between the observations and the OGCMs (Figure 3). The models suggest that the subsurface current is not persistent across the years during fall and winter (Movie S1). Longitude-depth transects of meridional velocity across 8° N from POP and HYCOM (not shown) do not show a well-defined undercurrent near the east coast of Sri Lanka in the top 1000 m of the water column during these seasons.

### 3.3 Depth-integrated volume transport

Monthly volume transports computed from the OGCMs and volume transport computed from the discrete glider sections (referenced to depth-average velocity; e.g. Figure 2) across the 8° N transect between the eastern coast of Sri Lanka and 82.5° E (Figure 1a) over the upper (0-200 m) and lower (200-1000 m) layers are presented in Figure 5. The seasonal variation from HYCOM, POP, and glider measurements agree well, especially in the subsurface layer. Mean volume transport in the upper layer (0-200 m) has a semi-annual cycle that ranges from 1 to 6 Sv into and out of the BoB (Figure 5a). Southward flows are observed in the surface layer during boreal summer and winter. The flows are weakly northward during boreal spring and fall.

Depth-integrated transport in the subsurface layer is southward throughout the year except during the summer when the northward-flowing undercurrent is present (Figures 3, 5b). The undercurrent transport is approximately 5-7 Sv out of and 3-11 Sv into the BoB during boreal spring and summer, respectively (Table 3). Moreover, the OGCMs suggest only a small mean volume transport of the subsurface current

during fall and winter compared to its standard deviation (Figure 5b) implying that the fall and winter undercurrent is not well defined in these seasons and indeed can flow northward in some years. Volume transport over the 0-1000 m layer ranges from -12 to 5 Sv and exhibits an annual cycle similar to that in the lower (200-1000 m) layer (Figure 5c).

## 4 Discussion

The circulation in the southern BoB is complex as it is controlled by various forcings, such as local winds, Ekman pumping, and equatorial waves, which impact the region in different seasons. The mechanisms driving the undercurrent remain unclear and although beyond the scope of this study we put forward some informed hypotheses based on previous studies and our observations. The Rossby waves, equatorial waves, and Ekman pumping are likely to be important for producing the undercurrent: a Hovmöller diagram of POP meridional velocity across 8° N (Figure 6) shows the westward propagation of the velocity signal at 729 m depth originating from the eastern boundary of the BoB that takes about four months to cross the southern BoB. It has a propagation speed of 12 cm s$^{-1}$ in good agreement with the phase speed of a mode 2 baroclinic Rossby wave in this region (Shankar et al., 1996; Killworth and Blundell, 2003; Wijesekera et al., 2016a). A westward-propagating signal first develops at the eastern boundary of the BoB in April-May at the same time that the Wyrtki Jet, a surface-intensified eastward-flowing current observed in the equatorial Indian Ocean during the monsoon transitions (Wyrtki, 1973), reaches this eastern boundary. The westward-propagating signal reaches the east coast of Sri Lanka in September contributing to the southward-flowing subsurface current (Figures 3); this is consistent with findings from the LSM study (McCreary et al., 1996). In April, the westward propagating meridional flows originating at 83° E-85° E (Figure 6b) are associated with the subsurface anticyclonic eddy observed along the east coast of Sri Lanka during the summer (Figure 4b). The influence of equatorial waves on the undercurrent is still unclear and more studies are needed. Upward phase propagation is present during summer and early fall (Figure 3) implying the influence of equatorial waves on the subsurface flow during this time (Luyten and Roemmick, 1982). Equatorial waves are also known to impact currents along the Indian east coast (Mukherjee et al., 2014) and Sri Lankan south coast (Schott et al., 1994; Shankar et al., 2002).

Ekman pumping can also significantly impact the circulation along the eastern coast of Sri Lanka (McCreary et al., 1996; Vinayachandran et al., 1999). The LSM study of McCreary et al. (1996) suggested that Ekman pumping drives the undercurrent east of Sri Lanka from April to December. The model undercurrent has a core at 400-500 m that can reach a maximum speed of 10 cm s$^{-1}$ southward and northward in March-April and August, respectively. Although this value agrees quite well with that of the spring undercurrent observed in the observations and OGCMs, the magnitude given by the LSM during the summer is about half that shown in this study (Figure 3). During the winter, the mean subsurface current of the observations and OGCMS agrees with the results from the LSM (McCreary et al., 1996) within one standard deviation. This implies that a combination of local alongshore winds, Ekman pumping, and equatorial waves, which are the main mechanisms driving the subsurface current during the winter in the LSM, is important in producing the subsurface current in only some years.

Circulation in the subsurface layer along the Sri Lankan east coast is characterized by reversed flows relative to the surface during boreal spring and summer similar to the subsurface circulation along the Indian east coast and Sri Lankan south coast described by previous studies (Schott et al., 1994; McCreary et al., 1996; Mukherjee et al., 2014). During the southwest monsoon, the current along the Sri Lankan south coast (Schott et al., 1994), Sri Lankan east coast (Figure 2d-f; Table 2), and Indian east coast (Mukherjee et al., 2014) reverses its direction below approximately 100-150 m. The subsurface current flows eastward along the southern coast of Sri Lanka (Schott et al., 1994) in agreement with the circulation pattern derived from the POP velocity fields (Figure 4b) and poleward along the Sri Lankan and the Indian east coast (McCreary et al., 1996; Mukherjee et al., 2014). This suggests the possibility of a subsurface conduit connecting the region off the Sri Lankan south coast to the northern BoB. However, the subsurface poleward current along the Indian east coast is not always apparent (Mukherjee et al., 2014), although it is also possible that the undercurrent during this season occurs below the deepest measurement at 300 m from the Mukherjee et al. (2014) study. In boreal spring, mooring measurements along the Sri Lankan south coast (Schott et al., 1994) verifies the existence of the eastward-flowing subsurface current observed in the POP velocity fields (Figure 4a). The subsurface current extends from approximately 250 m to 1010 m (Schott et al., 1994). Unlike the summer subsurface circulation, flow at and north of 12° N is poleward over 0-300 m (Mukherjee et al., 2014), in the opposite direction to the undercurrent off the Sri Lankan east coast (Figures 2a-c, 3). However, since the core location of the spring undercurrent is highly variable from year to year, direct velocity measurements in the deeper layer would help to gain a better understanding of the pathways of the subsurface circulation along the western boundary of the BoB during boreal spring.

Although velocity and salinity fields from POP suggest the possibility of significant salt exchange between the AS and the BoB, the estimated salt transport via the undercurrent is small based on the data and models we have available. POP velocity fields show that the summer undercurrent injects relatively saline water into the western boundary of the BoB that can be trapped in a northward-propagating seasonal anticyclonic eddy (Figure 4b). The source of the saline water is mostly from the southwest of Sri Lanka, although sometimes, such as in 1998, it originates from the eastern BoB (Movie S1). Relatively fresher water is transported southward along the east coast of Sri Lanka in the subsurface layer in spring (Figures 4a, 7). It can be advected westward along the Sri Lankan south coast in some years depending on the strength and location of the deep eastward-flowing Wyrtki jet (Reppin et al., 1999; Movie S1). The time series of salinity averaged over the width of the undercurrent (from the east coast of Sri Lanka to 82.5° E) and the volume transport of the POP undercurrent (over 200-1000 m) along 8° N clearly show a positive correlation *i.e.* the undercurrent tends to export freshwater from the BoB and import saline water into the BoB (Figure 7). The hydrography-Argo profiles, glider measurements, and HYCOM also show that poleward subsurface transport is associated with relatively saline water and vice versa (not shown). The correlation coefficient between the subsurface transport and salinity computed from POP is 0.50, significant at the 95% confidence level. The salt flux due to the fluctuations of the flow and of the salinity (departures from the 13-year climatology) in the 200-1000 m layer from POP is $0.04 \times 10^6$ kg s$^{-1}$ (Figure 7), while the estimated salt flux from the hydrography-Argo profiles and glider measurements are $0.19 \times 10^6$ and $0.09 \times 10^6$ kg s$^{-1}$, respectively.

The difference between the observational and POP salt fluxes arises from the smaller salinity range of the model compared to the observations (Table 3). The estimated salt input is up to 4% of the total expected salt transport into the BoB of $4.5\text{-}4.9\times10^6$ kg s$^{-1}$, which is estimated from the annual freshwater input of 0.13-0.14 Sv (Rao and Sivakumar, 2003; Sengupta et al., 2006; Wilson and Riser, 2016). Note that the total net salt transported by POP into the BoB across 8° N is $4.3\times10^6$ kg s$^{-1}$; this value is in good agreement with the expected salt transport discussed above. Estimates of salt flux by the *mean* flow over the same depth layer from POP and HYCOM are even smaller: the mean undercurrent transport of 1.5-2 Sv in the OGCMs is fresher than the interior northward flow as judged from HYCOM and sparse hydrography and Argo profiles (not shown), but the salinity contrast at this depth layer is smaller than 0.01. In addition, the small salt transport from the mean flow in the 200-1000 m layer might be due to the core of the undercurrent moving vertically such that this layer may sometimes include opposite flows or smaller salinity contrasts with the interior (Figures 2, 3; Table 1, 2).

An alternative interbasin-exchange pathway is through the interior of the BoB, particularly during the southwest and northeast monsoon (Vinayachandran et al., 1999; Wijesekera et al., 2015; Wijesekera et al., 2016b). Observations indicate that the eastward-flowing southwest monsoon current (SMC) has a role in the injection of saline water originating in the AS into the southern central BoB during early summer. As the summer progresses, the seasonal cyclonic (*i.e.* the SLD) and anticyclonic eddies influence the pathway of the SMC (Vinayachandran and Yamagata, 1997; Vinayachandran et al., 1999). Mooring observations at 85.5° E from 5° N to 8° N reveal a northward-flowing subsurface SMC that is associated with high salinity water (Wijesekera et al., 2016a; Wijesekera et al., 2016b). During the northwest monsoon, mass and salt exchange between the AS and the BoB is also observed in the interior over the 50-75 m layer in the southern BoB, approximately between 82.5° E and 85 °E (Wijesekera et al., 2015).

# 5 Summary

The velocity fields derived from hydrography and Argo profiles, glider measurements, POP, and HYCOM reveal the presence of an undercurrent off the east coast of Sri Lanka during boreal spring and summer, which has opposite direction to the seasonally reversing surface flow. The location and width of the core of the undercurrent change seasonally. The undercurrent shows the greatest interannual variability during spring, which is likely due to the influence of El Niño events. The observations and OGCMs suggest that the undercurrent over the 200-1000 m layer transports more water than the surface current over the 0-200 m layer. The mechanisms driving the undercurrent are still unclear, though Ekman pumping is likely to affect the undercurrent especially during the boreal spring (McCreary et al., 1996; Vinayachandran et al., 1999). Upward phase propagation is distinct during the summer suggesting the influence of equatorial forcing on the circulation across 8° N. Our analyses do not observe the direct influence of Rossby waves on the modification of the spring and summer undercurrent. Although the POP velocity fields suggest a potential pathway in salt exchange between the AS and the BoB, salt transport across 8° N by the undercurrent estimated from POP and the observations is expected to be less than 4% of the total salt input into the BoB required to balance the

freshwater sources. More studies are needed to determine the mechanisms driving the undercurrent and its role in interbasin salt and mass exchange. In turn, this knowledge will lead to a better understanding of property exchanges in the northern Indian Ocean.

**Acknowledgements**

This work is supported by the US Office of Naval Research (ONR) as part of two ONR Departmental Research Initiatives: Air-Sea Interactions Regional Initiative (ASIRI) and Northern Arabian Sea Circulation – autonomous research (NASCar) projects through ONR grants N00014-14-1-0629 (A. Anutaliya and U. Send), N00014-15-1-2189 (J. L. McClean), N00014-16-1-2313 (J. Sprintall), N00014-13-1-0478, N00014-15-1-2296, N00014-15-1-2231 (L. Rainville and C. Lee). J. Metzger and A.
Wallcraft are supported by the "Eddy resolving global ocean prediction including tides" project under ONR program element 0602435N. Hydrographic profiles are from the National Center of Environmental Information (Boyer et al., 2013) (https://www.nodc.noaa.gov/access/index.html). Argo data were collected and made freely available by the International Argo Program and the national programs that contribute to it (http://www.argo.ucsd.edu, http://argo.jcommops.org). The Argo Program
is part of the Global Ocean Observing System. The altimeter products were produced by Ssalto/Duacs and distributed Aviso, with support from Cnes (http://www.aviso.altimetry.fr/duacs/).  The HYCOM simulation was performed using grants of computer time from the Department of Defense High Performance Computing Modernization Program. This is NRL contribution NRL/JA/7320-17-3395, which is approved for public release and distribution is unlimited. The POP simulation was carried
out using the Extreme Science and Engineering Discovery Environment (XSEDE), which is supported by National Science Foundation grant number ACI-1548562. POP output is available from XSEDE. Thanks to He Wang (SIO) for helpful discussions about the salt transport calculation. Also, thanks to Judy Gaukel (SIO) for extracting and transferring the OGCM files.

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

**Tables and Figure**

| | $v_{max}$ (cm s$^{-1}$) | Depth of $v_{max}$ (m) | Position of $v_{max}$ (° E) | Depth of reversal (m) | Width (km) |
|---|---|---|---|---|---|
| Glider | -25.3 | 1000.0 | 82.06 | 232.0 | 107.8 |
| HYCOM | -16.2 | 570.4 | 82.00 | 83.3 | 151.9 |
| POP | -18.9 | 268.5 | 81.95 | 77.4 | 209.1 |

**Table 1. Statistics of the undercurrent across 8° N in March from the glider geostrophic velocity as presented in Figure 2 and mean absolute velocity from HYCOM and POP. Here, $v_{max}$ represents maximum meridional velocity; depth of reversal is defined as the depth of the zero meridional velocity at the location of the maximum velocity; width of the undercurrent is defined as the distance from the coast at the depth of maximum velocity to the location where the meridional velocity is zero.**

| | $v_{max}$ (cm s$^{-1}$) | Depth of $v_{max}$ (m) | Position of $v_{max}$ (° E) | Depth of reversal (m) | Width (km) |
|---|---|---|---|---|---|
| Glider | 29.6 | 1000.0 | 81.90-82.40 | 117.3-300 | 206.9 |
| HYCOM | 20.6 | 800.8 | 82.08 | 244.8 | 165.1 |
| POP | 27.6 | 918.4 | 82.15 | 244.1 | 172.8 |

**Table 2. Same as Table 1 but for June.**

| | Mean transport (Sv) | | Seasonal salinity range |
|---|---|---|---|
| | Spring | Summer | |
| Hydrography -Argo | -6.9 | 9.7 | 0.09 |
| Glider | -5.7 | 11.42 | 0.04 |
| HYCOM | -4.6 | 3.0 | 0.01 |
| POP | -5.7 | 6.8 | 0.01 |

**Table 3. Mean transport and seasonal salinity range of the undercurrent (200-1000 m) across 8° N in boreal spring (February to mid-April as highlighted by white lines in Figure 3) and summer (June to mid-August as highlighted by yellow lines in Figure 3) from hydrography-Argo profiles, glider measurements, HYCOM, and POP. Seasonal salinity range is the approximate difference between boreal spring and summer extremes in average seasonal cycle. Note that the statistics representing glider measurements are derived from gridded velocity referenced to its depth-averaged velocity (e.g. Figure 2) and salinity interpolated onto 8° N.**

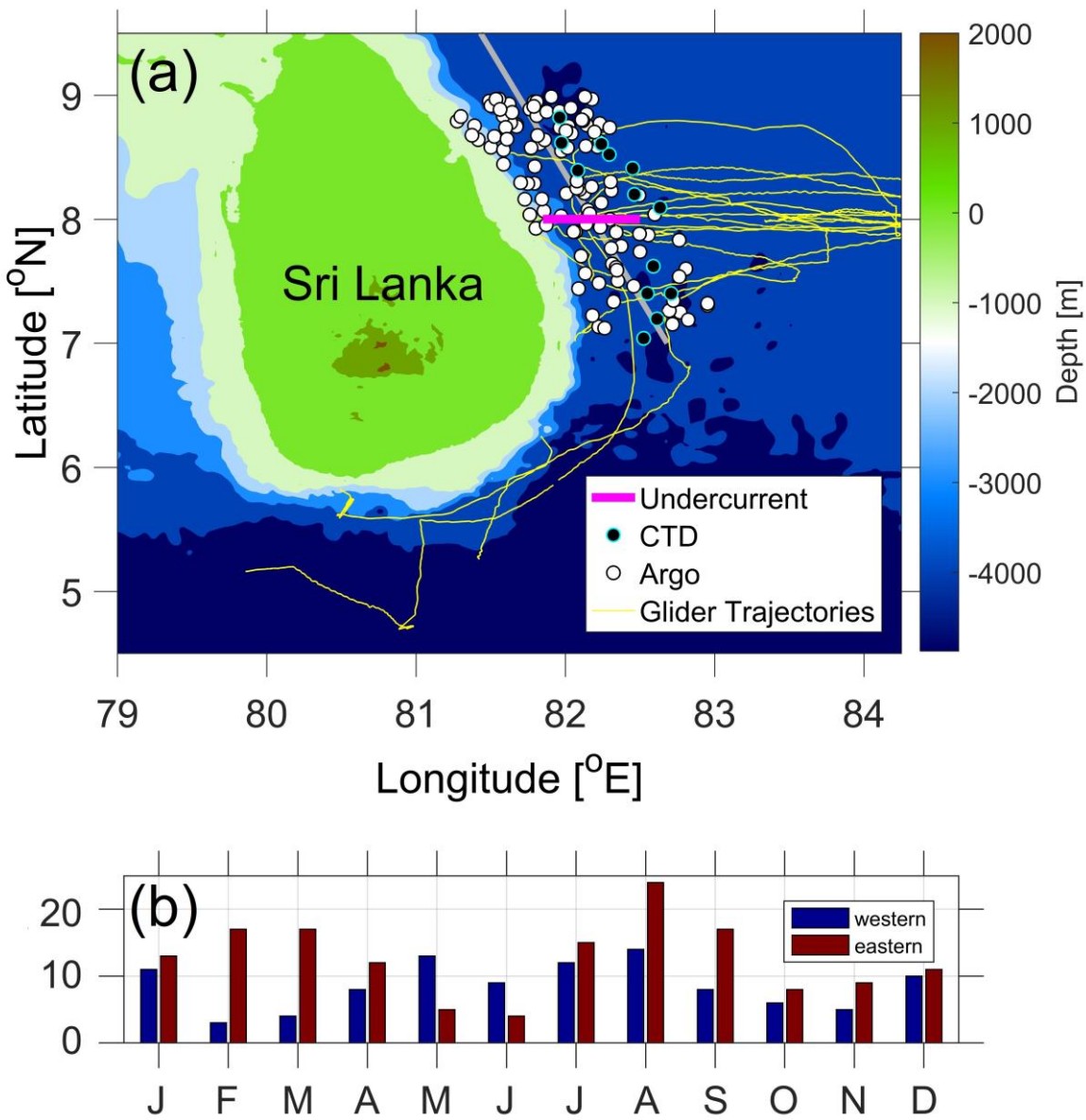

**Figure 1: Bathymetry around Sri Lanka (a) with the location of the shipboard hydrography measurements (black dots), Argo profiles (white dots), and glider measurement transects (yellow lines). The grey line divides hydrography and Argo measurements into the eastern and western sectors used in the seasonal mean dynamic height calculation. The OGCM outputs are interpolated along 8° N (magenta line). The number of hydrography and Argo profiles available in the mean dynamic height calculation for each month for the eastern (red) and western (blue) sectors are shown in (b).**

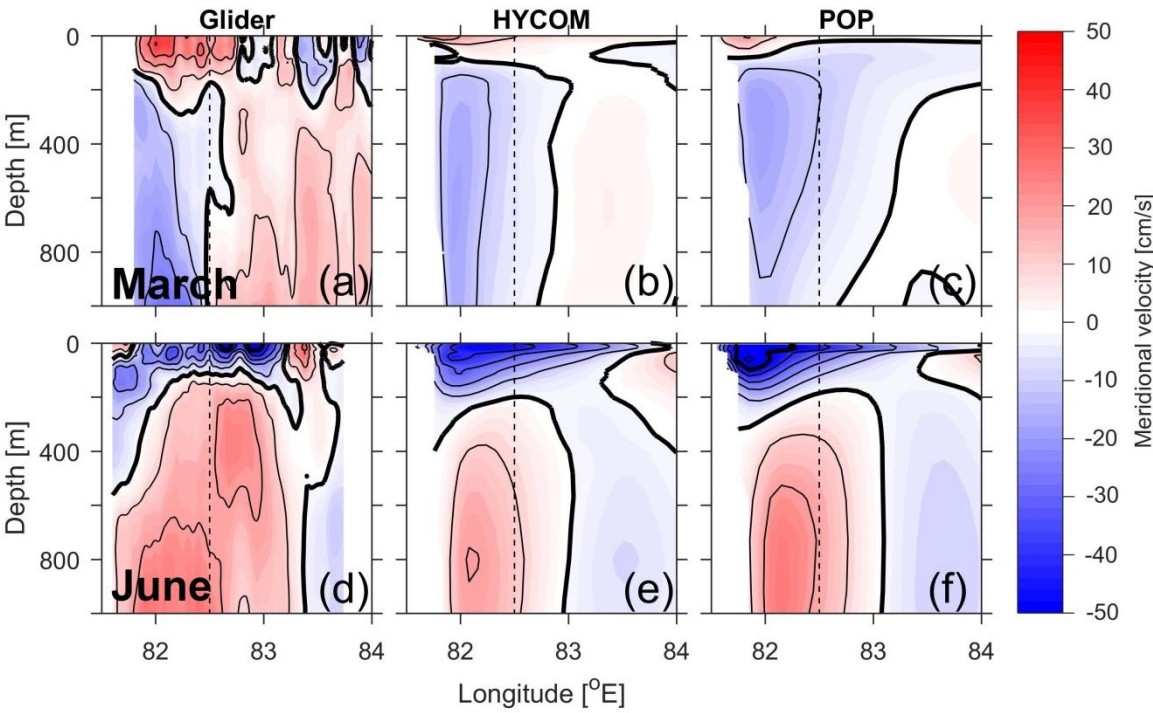

**Figure 2: Geostrophic meridional velocity referenced to the depth-averaged velocity across 8° N from glider measurements during March 18-30, 2014 (a) and May 24 – Jun 8, 2014 (d), the mean absolute meridional velocity from HYCOM (b and e) and POP (c and f) in March and June over 2003-2012 and 1995-2007 periods, respectively. The black dashed line marks the typical eastern extent of the undercurrent core; the region bounded to its west is used for a subsequent averaging in Figure 3. Thin black lines are plotted every 10 cm s$^{-1}$, and thick black lines are plotted every 50 cm s$^{-1}$.**

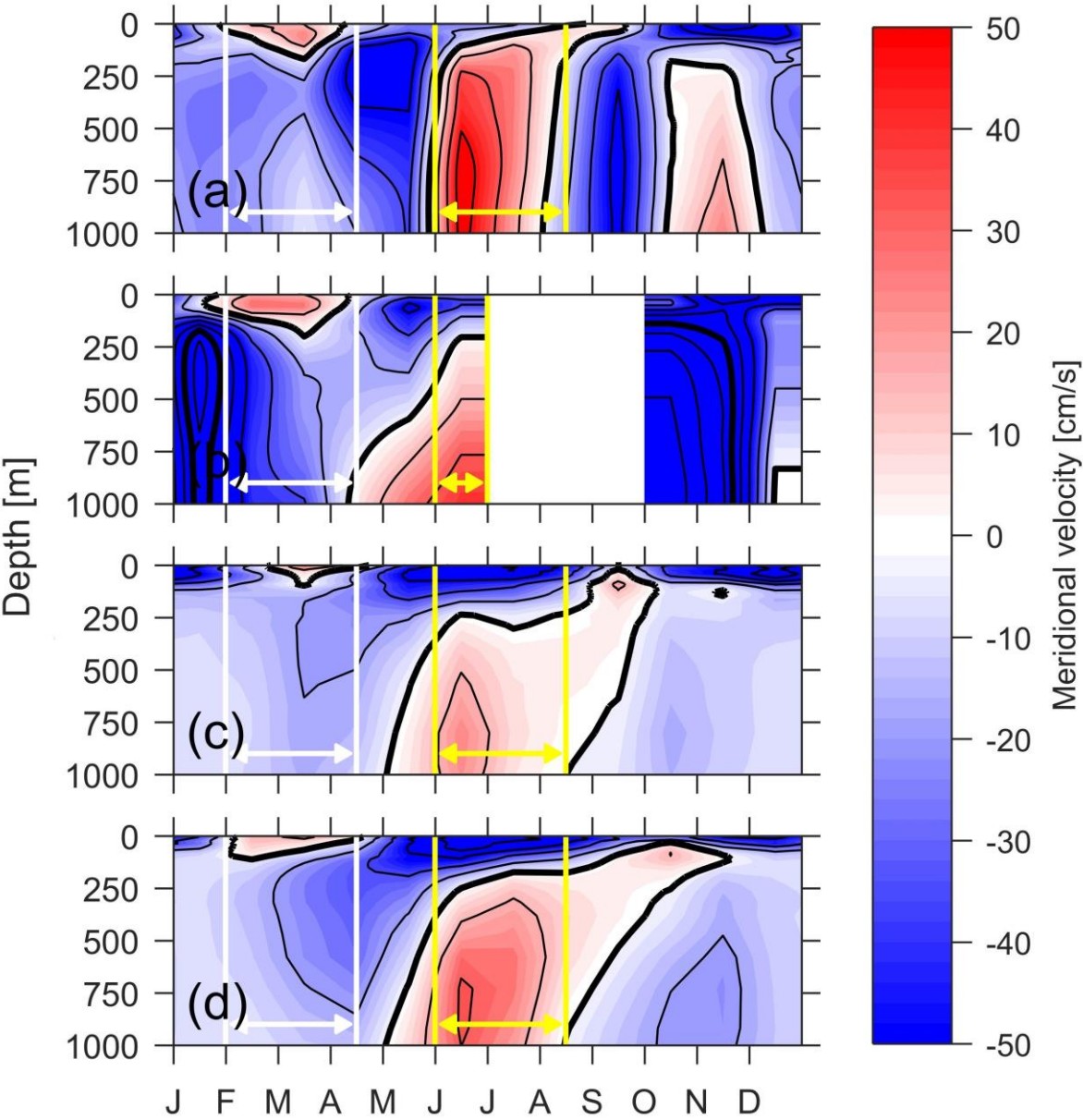

**Figure 3. Bimonthly mean seasonal variation of the meridional geostrophic velocity referenced to the surface satellite ADT across the magenta transect in Figure 1 from hydrography and Argo measurements (a), glider measurements (b), monthly mean meridional velocity profiles from HYCOM (c), and POP (d) output. Line contours are plotted in the same manner as in Figure 2. White and yellow lines indicate periods when the undercurrent is most pronounced.**

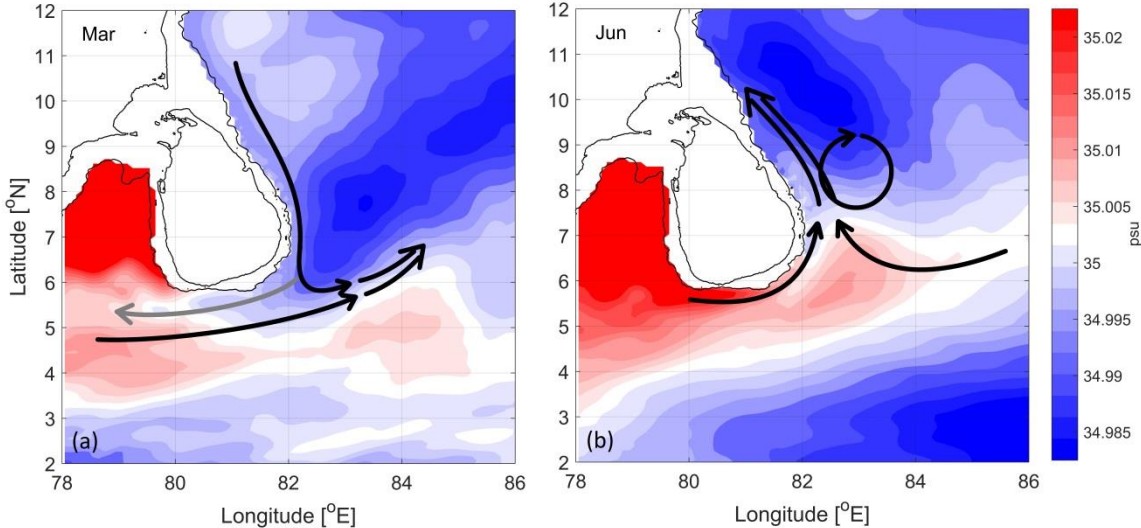

**Figure 4. Simplified schematic of the undercurrent along the east and south coast of Sri Lanka at 729 m depth based on POP outputs in March (a) and June (b) plotted over the corresponding POP monthly mean salinity. The black arrows represent the annual current patterns at 729 m, and the gray arrow shows circulation that is observed occasionally. The 0 and 1000 m bathymetry contours are plotted as thin black lines.**

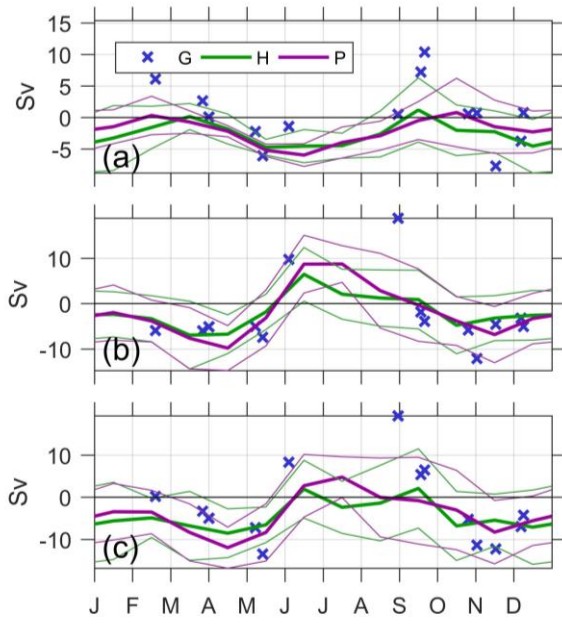

**Figure 5. Meridional volume transport across 8° N from the Sri Lankan east coast to 82.5° E (dashed line in Figure 2) calculated from glider geostrophic velocity referenced to the depth-averaged velocity (G; blue cross) and absolute velocity fields from HYCOM (H; thick green line) and POP (P; thick purple line) over the 0-200 m (a), 200-1000 m (b), and 0-1000 m (c) layers. Thin lines designate the mean value +/- one standard deviation.**

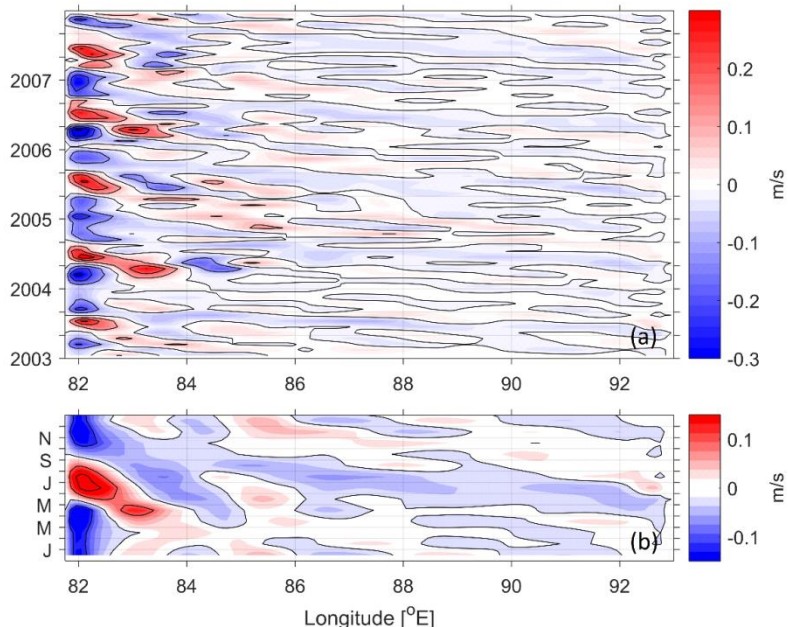

**Figure 6. Time-longitude plots of monthly meridional velocity (a) and seasonal mean velocity (b) over 2003-2007 across 8° N at 729 m from POP. Note the difference in color scale.**

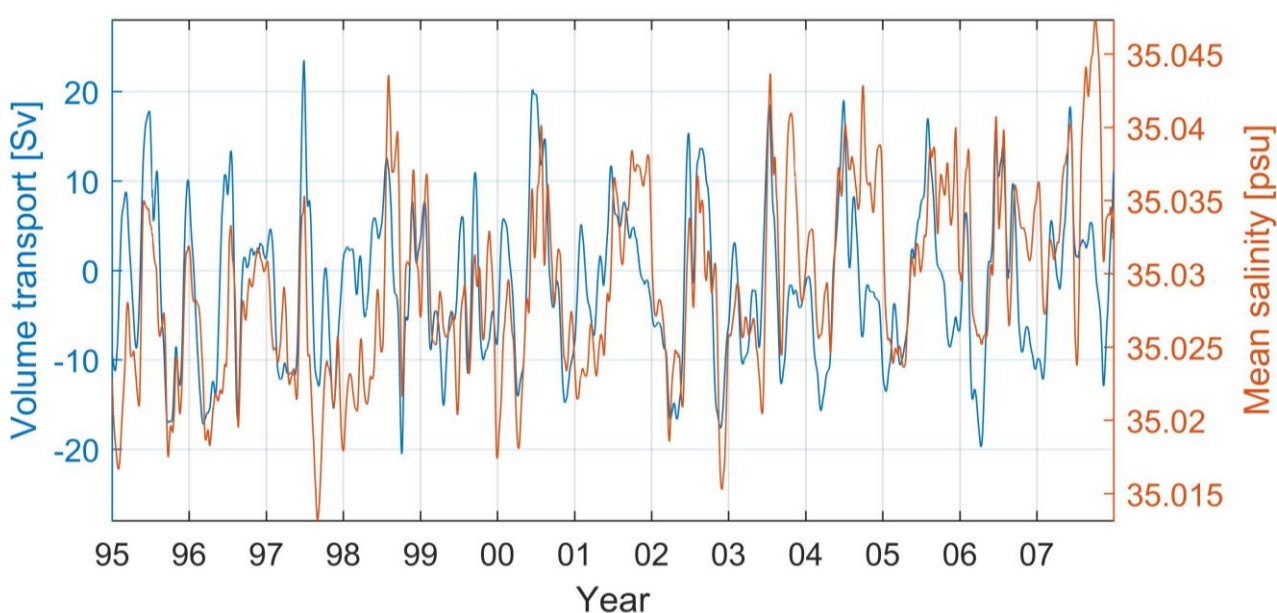

**Figure 7. Time series of volume transport (blue) and mean salinity (orange) between 81.5° E and 82.5° E over the 200-1000 m layer from POP.**

