# Peer review of "An Undercurrent off the East Coast of Sri Lanka"

_Ocean Science, 2017_

## Referee Comment (RC1) · Anonymous Referee #1 · 22 Aug 2017

Modelling studies have shown the existence of seasonal undercurrents along the east coast of Sri Lanka, and a few papers have explained the possible reasons that lead to the formation of these undercurrents. However, there were hardly any data available to validate the subsurface flow seen in these models. The present study describes the undercurrent off the east coast of Sri Lanka using both observations and models. The authors follow the data and method from Wijesekra 2015 and Lee 2016 to examine the characteristics of undercurrents, particularly at 8N. (The two papers focused on the surface circulation along the east coast of Sri Lanka.)

The highlight of this manuscript is the description of the undercurrents for the first time using available observational data. The authors further examine the spatial variation of the undercurrent using the model data.

[Figure]

Specific Comments:

1. P4 L1-12. The level of no motion for the calculation of geostrophic velocity is not very clear here. If the level of no motion was picked at 1000 m, where the currents are relatively strong, there would be a bias in the geostrophic calculations. The motion would be more relative and not absolute.

2. Page 4, L2. Though the structures are in good agreement, there are differences in the intensity. For example, the offshore currents are very weak in the model, especially during March.

3. P4, L12. The undercurrents are confined to west of 82.5E only for the month of March. During June, it extends till 83.5E. That should be roughly 180 km wide. I would say that the under currents are broader during summer compared to spring. Also, why is the width of undercurrents kept as 140 km in the abstract? The value is never mentioned in the text anywhere. Same goes for the maximum speed of 45 cm/s.

4. P4, L35. The strongest southward undercurrent during March-April is not very evident in the observations, especially in the glider data. For the CTD-Argo data, there is no evident undercurrent in April, when the strongest sub-surface flow is observed.

5. P4, L35. In the previous line, it is mentioned that the core of the undercurrent is observed at around 300 m. All of a sudden, without any justification, the undercurrents at 729 m is shown to highlight the spatial circulation. As the core of undercurrent changes from season to season, a decent justification should be provided to pick the 729 m depth. How different is the circulation at 729 m from that of 300 m?

6. P5, L1-11. The evidence of upward phase propagation (generally implies remote forcing) is not highlighted here. The phase propagation is possibly caused by interior Ekman pumping (McCreary et al. 1996, Fig 6c), and is not observed during spring. Note that the propagation is not evident in the CTD/Argo data though.

7. P6, L19-22. Why is the statistics presented only for fall and winter? Please show for

other seasons too and tabulate them.

8. P6, L1. How does undercurrent play an important role in salt exchange, when it is mentioned later in the text that its contribution is only 1% of the total salt transport in BoB? This is even mentioned in the abstract. The statement should be made more clear.

9. P7 1-12. What is the surface current transport, undercurrent current transport, and the total transport at 8N? Please quantify them. Do undercurrents roughly follow the total transport or not? Try replicating Figure 4 of McCreary et al. (1996) at 8N. The ship drift velocity could be replaced with the undercurrent velocity or transport. I'm curious. Because, the transport reverses during the summer monsoon along the east coast of India and not the surface circulation (McCreary, 1996). Therefore, the surface currents have a strong annual cycle, whereas the transport has a semiannual cycle. Is the reverse true here?

10. P7 14-25. Why is the comparison made between Mindanao undercurrent (which in the Pacific Ocean)? Undercurrents are present all around the world. A general comparison would have made more sense. I would have preferred a comparison with the undercurrents observed along the east coast of India. The dynamics of both these regions are interlinked. For example, direct current measurements show that the seasonal undercurrents are not very prominent (in the top 300 m) north of 12N (See Mukherjee et al. 2014) along the east coast of India. The undercurrents at 12N are evident at a shallower depth and are present because of strong upward phase propagation. A similar comparison with earlier hydrographic observations could also be made where the transport estimates are available.

11. Theoretical studies have shown a presence of undercurrents during the winter (McCreary et al. 1996). This difference has not been highlighted well in the study. The absence of the winter undercurrent leads to an annual cycle, which is in contrast to surface currents, where the semi-annual component is prominent. Note that the winter

undercurrent is present in the CTD-Argo data implying the presence of semi-annual cycle during some years. This point is not mentioned in the text.

12. It would be good to summarize the characteristics of undercurrents in a table. For example, you could divide the columns into seasons and mention the transport estimates, maximum current velocity, depth of the maximum velocity, and their necessary statistics (from both observations and model).

13. The discussion section, in general, could be improved.

The authors should further note that currents along the coast of Sri Lanka are complex. The complexity arises because the contribution from each of the forcing varies from year to year. For example, the climatology of alongshore currents from ship drift data show equatorward flow south of 8N (Mukherjee et al 2014), and the satellite data shows an annual cycle both south and north of 8N (Mukherjee et al 2014, Lee et al 2016). It's around 8N that the semi-annual cycle is more evident and this location may not represent the actual circulation along the entire Sri Lankan east coast.
* * *

---

## Referee Comment (RC2) · Anonymous Referee #2 · 23 Aug 2017

In the manuscript authors describe surface and sub-surface current along 8°N off east coast of Sri Lanka. They use observational data from CTD, ARGOS and gliders and the HYCOM and POP model outputs to describe the current structure at seasonal scales.

**Comments**

The literature review of the authors is inadequate. Many of the papers which talk about under currents around  Sri Lanka are not mentioned.

For example, Schott et al., JGR, 1994 which talks about opposite directions of surface and sub-surface currents. K1 mooring at 5.5N shows opposite currents during July-Aug.

Wijesekera et al., JPO 2016 uses ADCP data to describe the surface and sub-surface currents at 5-8°N  off east coast of Sri Lanka.

Shankar et al., 2002 PIO  is not mentioned, which talks about currents at 8N.

**Page No. 4 lines 23-25:**  The surface current reverses its direction four times a year ....

**Comment:** I do not think it is a correct statement. Shetye et al., 1996 talks about EICC during northeast monsoon. McCreary et al., 1996 talks about two times change in EICC direction, Durand et al., 2009 does not mention about four times change in the current direction.

It would be better if authors can draw a schematic diagram to represent the EICC directions as studied by the aforementioned papers and compare it with present study.

**Page No. 6 lines 22, 27:** A westward -propagating southward current....
The westward propagating northward and southward flows...

**Comment:** What do authors means by that?  It is totally confusing. A similar Hovmoller diagram is shown in Shankar et al., 2002. Authors can read that and describe this figure in a better way.

**Page 6 Lines 6-8:** The undercurrent potentially plays an important role in salt and mass exchange between the AS and the BoB..

**Comment:** But the movie shows that the dominant source of saline water is from eastern BoB. Very little contribution is from AS

**Page 6 Lines 30-35:** The stratified linear model study (McCreary et al., 1996) also indicates the role of ekman pumping ....

**Comment:** In McCreary et al., 1996 paper, at 8N, Equatorial forcing and Ekman pumping both look important for currents below 200m, though the current magnitude is very weak in both cases. The authors only say that ekman pumping is important without a convincing argument.

**Page 6 Lines 30-35:** The last paragraph of the discussion section, deals with currents in Pacific Ocean.

**Comment:** This seems completely out of context.

Authors' knowledge of the dynamics of EICC is not convincing in this manuscript. The manuscript reads patchy and like a collection of bits and pieces.

**This manuscript needs a major revision and I do not recommend for publication in the present form.**

---

## Author Comment (AC1) · 13 Oct 2017

**Reviewer Responses**

An Undercurrent off the East Coast of Sri Lanka

Arachaporn Anutaliya et al.

The author comment is presented in the following sequence: (1) itemized comments from the referee in black, (2) author's response in red, (3) author's changes in manuscript in red italic.

**Anonymous Referee #1**

1. P4 L1-12. The level of no motion for the calculation of geostrophic velocity is not very clear here. If the level of no motion was picked at 1000 m, where the currents are relatively strong, there would be a bias in the geostrophic calculations. The motion would be more relative and not absolute.

   (2) The velocity from the glider presented in figure 2 is referenced to its depth-averaged velocity, while that in figure 3 is referenced to the surface velocity derived from satellite absolute dynamic topography. Velocities from POP and HYCOM in both figures are total velocity. We have rewritten the text to make this clearer.

   (3) Clarification has been added to the manuscript (P4, L13-17).
   *"As noted above, geostrophic velocities derived from glider measurements, during March 18-30, 2014 (Figure 2a) and May 28–June 8, 2014 (Figure 2d), are referenced to the depth-averaged velocity measured directly by the gliders. Velocity transects from HYCOM (Figure 2b, e) and POP (Figure 2c, f) were constructed from monthly averages of total velocity in March and June for the periods of 2003-2012 and 1995-2007, respectively."*

   *Figure caption of Fig. 2 is also revised.*
   *"Figure 2: Geostrophic meridional velocity referenced to the depth-averaged velocity across 8° N from glider measurements during March 18-30, 2014 (a) and May 24 – Jun 8, 2014 (d), the mean absolute meridional velocity from HYCOM (b and e) and POP (c and f) in March and June over 2003-2012 and 1995-2007 periods, respectively. The black dashed line marks the typical eastern extent of the undercurrent core; the region bounded to its west is used for a subsequent averaging in Figure 3. Thin black lines are plotted every 10 cm s$^{-1}$, and thick black lines are plotted every 50 cm s$^{-1}$."*

2. Page 4, L2. Though the structures are in good agreement, there are differences in the intensity. For example, the offshore currents are very weak in the model, especially during March.

   (2) The reviewer is correct, the intensity differs between the models and observations, as well as from season to season. The original text was not precise and has been made clearer. We have also added a table that describes the characteristics of the undercurrent in March (Table 1) and June (Table 2).

   (3) The manuscript has been revised (P4, L19-30).
   *"The model transects show reasonable agreement with that from the gliders in regard to the vertical structure of both the surface current and undercurrent. In March, the core of the observed undercurrent is located about 55 km from the*

*coast and has a maximum speed of 16-25 cm s$^{-1}$ southward (Table 1). HYCOM and POP have maximum flows in the reverse direction to the surface i.e. southward, at 570 and 268 m, respectively (Figure 2a-b) while glider measurements show maximum subsurface flow at the deepest measured depth of 1000 m (Figure 2c). In June, a northward undercurrent is observed below approximately 250 m with a maximum speed greater than 20 cm s$^{-1}$ at about 900 m (Figure 2d-f; Table 2). The summer undercurrent is approximately 165 km wide in the OGCMs. The velocity section derived from gliders shows a wider undercurrent of 210 km with two cores: a deep core that is approximately 100 km wide centered at 82.2° E and a shallow core at 400 m near 82.75° E (Figure 2d). This double-core feature is also evident in some individual model years, but is not present in the multi-year mean (Figure 2e, f)."*

*Tables have been added to the manuscript.*

| | $v_{max}$ (cm s$^{-1}$) | Depth of $v_{max}$ (m) | Position of $v_{max}$ (° E) | Depth of reversal (m) | Width (km) |
|---|---|---|---|---|---|
| Glider | -25.3 | 1000.0 | 82.06 | 232.0 | 107.8 |
| HYCOM | -16.2 | 570.4 | 82.00 | 83.3 | 151.9 |
| POP | -18.9 | 268.5 | 81.95 | 77.4 | 209.1 |

*Table 1. Statistics of the undercurrent across 8° N in March from the glider geostrophic velocity as presented in Figure 2 and mean absolute velocity from HYCOM and POP. Here, $v_{max}$ represents maximum meridional velocity; depth of reversal is defined as the depth of the zero meridional velocity at the location of the maximum velocity; width of the undercurrent is defined as the distance from the coast at the depth of maximum velocity to the location where the meridional velocity is zero.*

| | $v_{max}$ (cm s$^{-1}$) | Depth of $v_{max}$ (m) | Position of $v_{max}$ (° E) | Depth of reversal (m) | Width (km) |
|---|---|---|---|---|---|
| Glider | 29.6 | 1000.0 | 81.90-82.40 | 117.3-300 | 206.9 |
| HYCOM | 20.6 | 800.8 | 82.08 | 244.8 | 165.1 |
| POP | 27.6 | 918.4 | 82.15 | 244.1 | 172.8 |

*Table 2. Same as Table 1 but for June.*

3. P4, L12. The undercurrents are confined to west of 82.5E only for the month of March. During June, it extends till 83.5E. That should be roughly 180 km wide. I would say that the under currents are broader during summer compared to spring. Also, why is the width of undercurrents kept as 140 km in the abstract? The value is never mentioned in the text anywhere. Same goes for the maximum speed of 45 cm/s.

    (2) Width of the undercurrent varies seasonally, as does the depth and location of the velocity maximum core. Table 1 and 2 (above) have also been added to provide information about the characteristics of the undercurrent. The original text was not clear and we have revised the manuscript. The revised text and additional tables can be found in our response to Reviewer

Comment (2). In addition, we have removed the comment about the width of the undercurrent being 140 km in the abstract.

.

4. P4, L35. The strongest southward undercurrent during March-April is not very evident in the observations, especially in the glider data. For the CTD-Argo data, there is no evident undercurrent in April, when the strongest sub-surface flow is observed.

(2) The spring (March-April) undercurrent appears weaker in the observations (both CTD-Argo and glider data) compared to the OGCMs. Both depth and longitudinal position of the spring undercurrent core changes from year to year. This can mask the actual depth and longitude of the subsurface undercurrent in a sparsely sampled dataset such as the CTD-Argo profiles. We have made this point clearer in the revised manuscript. We have also included more recent Argo profiles into the calculation for Figure 3 and added arrows to each panel in the figure to make the presence of the undercurrent clearer to the reader.

*(3) The manuscript has been revised (P4, L35-39).*
*"The spring undercurrent exists in all years in the model simulations except for in 2007 and 2011 in HYCOM when an El Niño occurs in the preceding year. POP also shows a weakening of the spring undercurrent following an El Niño event, except in 1998 when the northward-flowing surface current is absent and the southward current extends from the surface to approximately 550 m depth."*

*The manuscript has been revised (P5, L30-37).*
*"An undercurrent flowing in the opposing direction to the surface current is a prominent and consistent feature among the observations and OGCMs in boreal spring (beginning of February to mid-April) and summer (beginning of June to mid-August) (Figure 3). In boreal spring, the southward-flowing undercurrent extends from approximately 200 to 900 m depth with a velocity maximum core in the range of 300-600 m depth. The velocities derived from hydrography and Argo profiles and glider measurements confirm the existence of the undercurrent below 150-250 m, although the magnitudes are not consistent (Figure 3). This may be due to interannual variation in intensity and position of the subsurface reversed flow, combined with the sporadic sampling of the hydrography and Argo data."*

[Figure]

*Figure 3. Bimonthly mean seasonal variation of the meridional geostrophic velocity referenced to the surface satellite ADT across the magenta transect in Figure 1 from hydrography and Argo measurements (a), glider measurements (b), monthly mean meridional velocity profiles from HYCOM (c), and POP (d) output. Line contours are plotted in the same manner as in Figure 2. White and yellow lines indicate periods when the undercurrent is most pronounced.*

5. P4, L35. In the previous line, it is mentioned that the core of the undercurrent is observed at around 300 m. All of a sudden, without any justification, the undercurrents at 729 m is shown to highlight the spatial circulation. As the core of undercurrent changes from season to season, a decent justification should be provided to pick the 729 m depth. How different is the circulation at 729 m from that of 300 m?

> (2) The depth of the undercurrent core fluctuates from season to season and from year to year. In boreal spring (March-April), the depth of the undercurrent is shallower (300-600 m), while the summer (June-July) maximum velocity core is at ~900 m. The POP depth level at 729 m is selected as a representative depth of the undercurrent during both seasons. Nonetheless, we realize that the definition of the depth level should have occurred earlier in the manuscript. We have revised the manuscript to better justify the choice of this depth level for the undercurrent. The circulation patterns in the POP model along the eastern and southern coasts of Sri

Lanka are very similar at 318 m and 729 m, although the current is stronger at 318 m.

*(3) The manuscript has been revised (P6, L4-19).*

*"To gain a better understanding of the subsurface circulation, we examine the POP velocity fields at 729 m (Movie S1), which encompasses the depth location of the undercurrent core in boreal spring and summer (Figures 2, 3; Tables 1, 2). In March, the undercurrent is strong and confined to 82.5° E along the entire Sri Lankan east coast (Movie S1). The undercurrent usually turns eastward around 6° N to combine with an eastward-flowing current along the Sri Lankan south coast, as shown in the simplified schematic derived from the POP 729 m velocity fields (Figure 4a). However, in some years (e.g. 1995, 1999, 2005, and 2007), the undercurrent turns westward around the southern coast of Sri Lanka and forms a narrow current, about 50-70 km wide, on the northern side of the eastward flow (Movie S1; Figure 4a). Note that a similar circulation is observed in the POP velocity fields at 318 m in March, albeit the flow is stronger. In boreal summer, there is a convergence of the eastward-flowing current along the south coast of Sri Lanka and a westward-flowing current in the southwestern BoB at approximately 7° N, 82.25° E (Movie S1), which is also shown in the simplified schematic (Figure 4b). Both OGCMs agree that the convergence produces the undercurrent flowing north along the east coast of Sri Lanka that injects relatively saline water into the BoB. In addition, an anticyclonic eddy often develops along the east coast, trapping local saline water inside, which is then propagated northward with the eddy (Movie S1)."*

6. P5, L1-11. The evidence of upward phase propagation (generally implies remote forcing) is not highlighted here. The phase propagation is possibly caused by interior Ekman pumping (McCreary et al. 1996, Fig 6c), and is not observed during spring. Note that the propagation is not evident in the CTD/Argo data though.

(2) As the reviewer notes, upward phase propagation, especially during the summer, is evident in both OGCM outputs. We have now added text in the Discussion of the manuscript describing how this upward phase propagation relates to the passage of planetary waves. We thank the reviewer for pointing out this oversight.

*(3) The manuscript has been revised (P5, L41).*

*"The OGCMs also suggest an upward phase propagation during summer and early fall (May- October) with a speed of approximately 1.5 m day$^{-1}$ in the upper 200 m of the water column (Figure 3c, d). This feature is also evident in the geostrophic velocity derived from hydrography-Argo profiles in the upper 100 m of the water column (Figure 3a)."*

*Discussion of the upward phase propagation has also been added (P7, L24-27).*

*"Upward phase propagation is present during summer and early fall (Figure 3) implying the influence of equatorial waves on the subsurface flow during this time (Luyten and Roemmick, 1982). Equatorial waves are also known to*

*impact currents along the Indian east coast (Mukherjee et al., 2014) and Sri Lankan south coast (Schott et al., 1994; Shankar et al., 2002)."*

7. P5, L19-22. Why is the statistics presented only for fall and winter? Please show for other seasons too and tabulate them.

(2) The statistics were only presented in winter in the original text to show that the winter subsurface flow is quite variable compared to its mean – this supports our suggestion that the winter undercurrent is not a permanent feature. However, we agree with the reviewer that the statistics should be presented for all seasons to highlight the existence of a spring and summer undercurrent and show that the fall and winter undercurrent are not permanent features. Hence, we have added Figure 5 (below) of the mean volume transport and the standard deviation computed from the OGCMs over the upper (0-200 m) and lower (200-1000 m) layer to show the size of the uncertainty of the undercurrent transport in each month compared to its mean. Volume transport computed from glider velocity referenced to its depth-averaged velocity in March and June when the undercurrent is distinct is also added to the figure for comparison.

(3) *A figure of volume transport in 0-200 m, 200-1000 m, and 0-1000 m layers has been added.*

[Figure]

*Figure 5. Meridional volume transport across 8° N from the Sri Lankan east coast to 82.5° E (dashed line in Figure 2) calculated from glider geostrophic velocity referenced to the depth-averaged velocity (G; blue cross) and absolute velocity fields from HYCOM (H; thick green line) and POP (P; thick purple line) over the 0-200 m (a), 200-1000 m (b), and 0-1000 m (c) layers. Thin lines designate the mean value +/- one standard deviation.*

8. P6, L1. How does undercurrent play an important role in salt exchange, when it is mentioned later in the text that its contribution is only 1% of the total salt transport in BoB? This is even mentioned in the abstract. The statement should be made more clear.

(2) As the reviewer notes, the role of the undercurrent in the salt exchange between the basins is still unclear. We have moderated the influence of the salt exchange attributed to the undercurrent in the revised text (Discussion found in our response to Reviewer Comment (13) and Abstract) so as to better reflect the uncertainty of the salt exchange.

(3) The manuscript has been revised (P8, L24- P9, L12).
"Although velocity and salinity fields from POP suggest the possibility of significant salt exchange between the AS and the BoB, the estimated salt transport via the undercurrent is small based on the data and models we have available. POP velocity fields show that the summer undercurrent injects relatively saline water into the western boundary of the BoB that can be trapped in a northward-propagating seasonal anticyclonic eddy (Figure 4b). The source of the saline water is mostly from the southwest of Sri Lanka, although sometimes, such as in 1998, it originates from the eastern BoB (Movie S1). Relatively fresher water is transported southward along the east coast of Sri Lanka in the subsurface layer in spring (Figures 4a, 7). It can be advected westward along the Sri Lankan south coast in some years depending on the strength and location of the deep eastward-flowing Wyrtki jet (Reppin et al., 1999; Movie S1). The time series of salinity averaged over the width of the undercurrent (from the east coast of Sri Lanka to 82.5° E) and the volume transport of the POP undercurrent (over 200-1000 m) along 8° N clearly show a positive correlation i.e. the undercurrent tends to export freshwater from the BoB and import saline water into the BoB (Figure 7). The hydrography-Argo profiles, glider measurements, and HYCOM also show that poleward subsurface transport is associated with relatively saline water and vice versa (not shown). The correlation coefficient between the subsurface transport and salinity computed from POP is 0.50, significant at the 95% confidence level. The salt flux due to the fluctuations of the flow and of the salinity (departures from the 13-year climatology) in the 200-1000 m layer from POP is $0.04 \times 10^6$ kg s$^{-1}$ (Figure 7), while the estimated salt flux from the hydrography-Argo profiles and glider measurements are $0.19 \times 10^6$ and $0.09 \times 10^6$ kg s$^{-1}$, respectively. The difference between the observational and POP salt fluxes arises from the smaller salinity range of the model compared to the observations (Table 3). The estimated salt input is up to 4% of the total expected salt transport into the BoB of $4.5-4.9 \times 10^6$ kg s$^{-1}$, which is estimated from the annual freshwater input of 0.13-0.14 Sv (Rao and Sivakumar, 2003; Sengupta et al., 2006; Wilson and Riser, 2016). Note that the total net salt transported by POP into the BoB across 8° N is $4.3 \times 10^6$ kg s$^{-1}$; this value is in good agreement with the expected salt transport discussed above. Estimates of salt flux by the mean flow over the same depth layer from POP and HYCOM are even smaller: the mean undercurrent transport of 1.5-2 Sv in the OGCMs is fresher than the interior northward flow as judged from HYCOM and sparse hydrography and Argo profiles (not shown), but the salinity contrast at this depth layer is smaller than 0.01. In addition, the small salt

*transport from the mean flow in the 200-1000 m layer might be due to the core of the undercurrent moving vertically such that this layer may sometimes include opposite flows or smaller salinity contrasts with the interior (Figures 2, 3; Table 1, 2)."*

*A table showing mean transport in boreal spring and summer and seasonal salinity range is added.*

| | Mean transport (Sv) | | Seasonal salinity range |
|---|---|---|---|
| | *Spring* | *Summer* | |
| *Hydrography -Argo* | *-6.9* | *9.7* | *0.09* |
| *Glider* | *-5.7* | *11.42* | *0.04* |
| *HYCOM* | *-4.6* | *3.0* | *0.01* |
| *POP* | *-5.7* | *6.8* | *0.01* |

*Table 3. Mean transport and seasonal salinity range of the undercurrent (200-1000 m) across 8° N in boreal spring (February to mid-April as highlighted by white lines in Figure 3) and summer (June to mid-August as highlighted by yellow lines in Figure 3) from hydrography-Argo profiles, glider measurements, HYCOM, and POP. Seasonal salinity range is the approximate difference between boreal spring and summer extremes in average seasonal cycle. Note that the statistics representing glider measurements are derived from gridded velocity referenced to its depth-averaged velocity (e.g. Figure 2) and salinity interpolated onto 8° N.*

*The abstract has been revised.*
*"The existence of a seasonally varying undercurrent along 8° N off the east coast of Sri Lanka is inferred from shipboard hydrography, Argo floats, glider measurements, and two Ocean General Circulation Model simulations. Together, they reveal an undercurrent below 100-200 m flowing in the opposite direction to the surface current that is most pronounced during boreal spring and summer and switches direction between these two seasons. The volume transport of the undercurrent (200-1000 m layer) can be more than 10 Sv in either direction, exceeding the transport of 1-6 Sv carried by the surface current (0-200 m layer). The undercurrent transports relatively fresh water southward during spring, while it advects more saline water northward along the east coast of Sri Lanka during summer. Although the undercurrent is potentially a pathway of salt exchange between the Arabian Sea and the Bay of Bengal, the observations and the OGCMs suggest that the salinity contrast between seasons and between the boundary current and interior is less than 0.09 in the subsurface layer, suggesting a small salt transport by the undercurrent of less than 4% of the salinity deficit in the Bay of Bengal."*

9. P7 1-12. What is the surface current transport, undercurrent current transport, and the total transport at 8N? Please quantify them. Do undercurrents roughly follow the total transport or not? Try replicating Figure 4 of McCreary et al. (1996) at 8N. The ship drift velocity could be replaced with the undercurrent velocity or transport. I'm curious.

Because, the transport reverses during the summer monsoon along the east coast of India and not the surface circulation (McCreary, 1996). Therefore, the surface currents have a strong annual cycle, whereas the transport has a semiannual cycle. Is the reverse true here?

(2) As noted above, we have added Figure 5 to the revised manuscript to show the change in transport over different layers and the total 0-1000m layer. The OCGMs show that the surface transport (0-200 m) reverses twice a year. The undercurrent (200-1000 m) has strong northward flow during the summer (June-July) and southward flow during the spring (March-April), both in the opposite direction to the surface current. During the winter, volume transport over the 200-1000 m depth is still southward but weak. Transport over the depth range 0-1000 m in the water column is most closely related to the subsurface structure. Unfortunately, this figure is difficult to compare to the volume transport plotted in Figure 4 of McCreary et al. (1996) as the transport is given relative to 1000 m, whereas the volume transport calculated in this paper is absolute transport. (The ship drift current only reflects the near surface current, whereas the focus of our paper is on the subsurface undercurrent).

(3) *Discussion on surface current transport, undercurrent transport, and the total transport have been analyzed and added to the text (P6, L30- P7, L4).*
*"Monthly volume transports computed from the OGCMs and volume transport computed from the discrete glider sections (referenced to depth-average velocity; e.g. Figure 2) across the 8° N transect between the eastern coast of Sri Lanka and 82.5° E (Figure 1a) over the upper (0-200 m) and lower (200-1000 m) layers are presented in Figure 5. The seasonal variation from HYCOM, POP, and glider measurements agree well, especially in the subsurface layer. Mean volume transport in the upper layer (0-200 m) has a semi-annual cycle that ranges from 1 to 6 Sv into and out of the BoB (Figure 5a). Southward flows are observed in the surface layer during boreal summer and winter. The flows are weakly northward during boreal spring and fall. Depth-integrated transport in the subsurface layer is southward throughout the year except during the summer when the northward-flowing undercurrent is present (Figures 3, 5b). The undercurrent transport is approximately 5-7 Sv out of and 3-11 Sv into the BoB during boreal spring and summer, respectively (Table 3). Moreover, the OGCMs suggest only a small mean volume transport of the subsurface current during fall and winter compared to its standard deviation (Figure 5b) implying that the fall and winter undercurrent is not well defined in these seasons and indeed can flow northward in some years. Volume transport over the 0-1000 m layer ranges from -12 to 5 Sv and exhibits an annual cycle similar to that in the lower (200-1000 m) layer (Figure 5c)."*

*More discussion is added to the discussion section of the manuscript (P7, L35-39).*
*"During the winter, the mean subsurface current of the observations and OGCMS agrees with the results from the LSM (McCreary et al., 1996) within*

*one standard deviation. This implies that a combination of local alongshore winds, Ekman pumping, and equatorial waves, which are the main mechanisms driving the subsurface current during the winter in the LSM, is important in producing the subsurface current in only some years."*

10. P7 14-25. Why is the comparison made between Mindanao undercurrent (which in the Pacific Ocean)? Undercurrents are present all around the world. A general comparison would have made more sense. I would have preferred a comparison with the undercurrents observed along the east coast of India. The dynamics of both these regions are interlinked. For example, direct current measurements show that the seasonal undercurrents are not very prominent (in the top 300 m) north of 12N (See Mukherjee et al. 2014) along the east coast of India. The undercurrents at 12N are evident at a shallower depth and are present because of strong upward phase propagation. A similar comparison with earlier hydrographic observations could also be made where the transport estimates are available.

(2) Our original thought was to compare and contrast the Sri Lanka undercurrent with other low latitude boundary currents, such as the Mindanao Current. However, we agree with the reviewer that a comparison with undercurrents observed in the BoB is more relevant. In the revised text the undercurrent off the Sri Lankan east coast is now discussed in the context of the undercurrents along the Indian east coast (the EICC- reported in Mukherjee et al., 2014) and the Sri Lankan south coast (SMC and WMC- reported in Schott et al., 1994). The comparison has been added to the discussion section found in our response to the Reviewer Comment (13).

*(3) The comparison has been added to the text (P8, L1-22).*
*"Circulation in the subsurface layer along the Sri Lankan east coast is characterized by reversed flows relative to the surface during boreal spring and summer similar to the subsurface circulation along the Indian east coast and Sri Lankan south coast described by previous studies (Schott et al., 1994; McCreary et al., 1996; Mukherjee et al., 2014). During the southwest monsoon, the current along the Sri Lankan south coast (Schott et al., 1994), Sri Lankan east coast (Figure 2d-f; Table 2), and Indian east coast (Mukherjee et al., 2014) reverses its direction below approximately 100-150 m. The subsurface current flows eastward along the southern coast of Sri Lanka (Schott et al., 1994) in agreement with the circulation pattern derived from the POP velocity fields (Figure 4b) and poleward along the Sri Lankan and the Indian east coast (McCreary et al., 1996; Mukherjee et al., 2014). This suggests the possibility of a subsurface conduit connecting the region off the Sri Lankan south coast to the northern BoB. However, the subsurface poleward current along the Indian east coast is not always apparent (Mukherjee et al., 2014), although it is also possible that the undercurrent during this season occurs below the deepest measurement at 300 m from the Mukherjee et al. (2014) study. In boreal spring, mooring measurements along the Sri Lankan south coast (Schott et al., 1994) verifies the existence of the eastward-flowing subsurface current observed in the POP velocity fields (Figure 4a). The subsurface current extends from approximately 250 m to*

*1010 m (Schott et al., 1994). Unlike the summer subsurface circulation, flow at and north of 12° N is poleward over 0-300 m (Mukherjee et al., 2014), in the opposite direction to the undercurrent off the Sri Lankan east coast (Figures 2a-c, 3). However, since the core location of the spring undercurrent is highly variable from year to year, direct velocity measurements in the deeper layer would help to gain a better understanding of the pathways of the subsurface circulation along the western boundary of the BoB during boreal spring."*

11. Theoretical studies have shown a presence of undercurrents during the winter (McCreary et al. 1996). This difference has not been highlighted well in the study. The absence of the winter undercurrent leads to an annual cycle, which is in contrast to surface currents, where the semi-annual component is prominent. Note that the winter undercurrent is present in the CTD-Argo data implying the presence of semi-annual cycle during some years. This point is not mentioned in the text.

   (2) The reviewer is correct, the difference between the mean undercurrent presented in the theoretical study (McCreary et al., 1996) and this study was not well highlighted. Our new Figure 5 presents the seasonal cycle of the mean transport in 0-200 m, 200-1000 m, and 0-1000 m. In the revised manuscript, we have discussed the difference between the findings of McCreary et al. (1996) and our study. Discussion of the seasonal cycle of the surface and subsurface current can be found in our response to Reviewer Comments (7) and (9).

12. It would be good to summarize the characteristics of undercurrents in a table. For example, you could divide the columns into seasons and mention the transport estimates, maximum current velocity, depth of the maximum velocity, and their necessary statistics (from both observations and model).

   (2) We thank the reviewer for the suggestion. We have added Tables 1 and 2 to summarize the characteristics of the undercurrent (see our response to Reviewer Comment (2)). Only the characteristics during March and June are summarized as the undercurrent is only prominent during these months and is not significant during fall and winter (Fig. 2 and 3 in the main text and Fig. 5).

13. The discussion section, in general, could be improved. The authors should further note that currents along the coast of Sri Lanka are complex. The complexity arises because the contribution from each of the forcing varies from year to year. For example, the climatology of alongshore currents from ship drift data show equatorward flow south of 8N (Mukherjee et al. 2014), and the satellite data shows an annual cycle both south and north of 8N (Mukherjee et al 2014, Lee et al 2016). It's around 8N that the semi-annual cycle is more evident and this location may not represent the actual circulation along the entire Sri Lankan east coast.

   (2) The reviewer is correct, the circulation across 8° N has a very complex structure as it represents interactions between different currents, such as the SMC/ WMC and the EICC, as well as the effect of strong wind stress curl and equatorial waves. All these phenomena vary on interannual time scales. We have revised our Discussion section to better recognize this complexity as suggested by the reviewer. In addition, we note that our new Figure 5a shows

the semi-annual signal in the surface layer that agrees with the ship drift data as pointed out by the reviewer. Nonetheless, the focus of our study is on the subsurface undercurrent and trying to better understand its characteristics.

(3) *The discussion section has been revised (P7, L5- P9, L24).*

[revised manuscript text omitted]

**Anonymous Referee #2**

1. The literature review of the authors is inadequate. Many of the papers which talk about under currents around Sri Lanka are not mentioned. For example, Schott et al., JGR, 1994 which talks about opposite directions of surface and sub-surface currents. K1 mooring at 5.5N shows opposite currents during July-Aug. Wijesekera et al., JPO 2016 uses ADCP data to describe the surface and subsurface currents at 5-8° N off east coast of Sri Lanka. Shankar et al., 2002 PIO is not mentioned, which talks about currents at 8N.

   (2) We thank the Reviewer for this suggestion and for the references. Our literature review has now been revised. We include a discussion on the undercurrent along the Indian east coast (EICC) and the Sri Lanka south coast (SMC/ WMC).

   *(3) The literature review has been revised (P1, L32 – P2, L31).*
   *"The surface current to the east of Sri Lanka is influenced by local alongshore winds, remote winds in the vicinity of the northern and eastern boundaries of the BoB, equatorial waves, and interior Ekman pumping (Yu et al., 1991; Shetye et al., 1993; McCreary et al., 1996; Shankar et al., 1996; Vinayachandran and Yamagata, 1997). This current has a strong seasonal pattern (Shankar et al., 2002; Lee et al., 2016) with recirculation loops that are highly variable in time and space (Durand et al., 2009). Some of the recirculations appear seasonally, such as the Sri Lanka Dome (SLD), a cyclonic eddy that is well developed in July (Vinayachandran and Yamagata, 1997). The SLD is driven by Rossby waves radiating from the eastern boundary and intensified Ekman pumping inside the BoB (Vinayachandran et al., 1999; Shankar et al., 2002; de Vos et al., 2014). The SLD propagates westward (Wijesekera et al., 2016b) toward the east coast of Sri Lanka resulting in a southward coastal surface flow during early summer. In October, the prevailing wind in the BoB starts reversing direction and blows southwestward. This marks the start of the northeast monsoon when the local wind along the east coast of India drives the East India Coastal Current (EICC) southward with speeds exceeding 1 m s$^{-1}$ extending from the east coast of India southward to the southern tip of Sri Lanka (Shetye et al., 1996; Wijesekera et al., 2015).*

   *The subsurface circulation in this region is also potentially important as high-salinity water from the AS can be subducted beneath the fresher surface water originating from river runoff and precipitation in the northern BoB (Rao & Sivakumar, 2003; Sengupta et al., 2006; Vinayachandran et al., 2013; Gordon et al., 2016; Jensen et al., 2016); this is evident in observations and model studies both during the northeast (Wijesekera et al., 2015) and southwest monsoons (June- August) (Wijesekera et al., 2016b). Little is known about the subsurface structure of the boundary current along the east coast of Sri Lanka. Mooring observations show a reversing subsurface current occurring off the southern coast of Sri Lanka; it is most distinct during*

*boreal spring and summer (Schott et al., 1994). In addition, reversal of the EICC along the east coast of India is observed below 100 m that is southward during the southwest monsoon and northward during the northeast monsoon, with the winter undercurrent being a more permanent feature (Mukherjee et al., 2014). The direction of this undercurrent is in good agreement with findings from a linear, continuously stratified ocean model (LSM) (McCreary et al., 1996). The LSM also suggests the presence of an undercurrent along the Sri Lankan east coast (centered at $8°N$) that reverses its direction twice a year with model speeds ranging from 6 cm s$^{-1}$ equatorward during boreal spring to 8 cm s$^{-1}$ poleward during summer. This undercurrent has not been observed before and will be the focus of this study. We will investigate the vertical structure and seasonal variability of subsurface flows in the boundary current system off the eastern coast of Sri Lanka using OGCMs as well as observations from shipboard hydrography, Argo floats, and glider measurements. Knowledge of the vertical structure, variability, and associated dynamics of the boundary current will contribute to a better understanding of mass and salt exchanges in the northern Indian Ocean."*

2. Page No. 4 lines 23-25: The surface current reverses its direction four times a year ….
Comment: I do not think it is a correct statement. Shetye et al., 1996 talks about EICC during northeast monsoon. McCreary et al., 1996 talks about two times change in EICC direction, Durand et al., 2009 does not mention about four times change in the current direction. It would be better if authors can draw a schematic diagram to represent the EICC directions as studied by the afore mentioned papers and compare it with present study.

   (2) The reviewer is correct and we have revised the statement

   (3) *The statement is corrected (P5, L13-14).*
   *"The surface current reverses its direction twice a year."*

3. Page No. 6 lines 22, 27: A westward -propagating southward current....
The westward propagating northward and southward flows...
Comment: What do authors means by that? It is totally confusing. A similar Hovmoller diagram is shown in Shankar et al., 2002. Authors can read that and describe this figure in a better way.

   (2) The statements describing the figure have now been revised to avoid confusion.

   (3) *The statement is revised (P7, L20-23).*
   *"In April, the westward propagating meridional flows originating at 83° E-85° E (Figure 6b) are associated with the subsurface anticyclonic eddy observed along the east coast of Sri Lanka during the summer (Figure 4b)."*

4. Page 6 Lines 6-8: The undercurrent potentially plays an important role in salt and mass exchange between the AS and the BoB.
Comment: But the movie shows that the dominant source of saline water is from eastern BoB. Very little contribution is from AS

   (2) The reviewer is correct that water from the AS is not the sole source and the eastern BoB is also likely to contribute saline water. However, the model shows that at 729 m depth, the majority of saline water from the eastern BoB

is transported westward along the Sri Lankan south coast. Only in a few years (for example 1998) is a coherent pattern not found between salinity along the southwestern and eastern coasts of Sri Lanka. The source of saline water advected by the undercurrent has been revised in the manuscript.

(3) *Clarification is added to the manuscript (P8, L26-29).*
*"POP velocity fields show that the summer undercurrent injects relatively saline water into the western boundary of the BoB that can be trapped in a northward-propagating seasonal anticyclonic eddy (Figure 4b). The source of the saline water is mostly from the southwest of Sri Lanka, although sometimes, such as in 1998, it originates from the eastern BoB (Movie S1)."*

5. Page 6 Lines 30-35: The stratified linear model study (McCreary et al., 1996) also indicates the role of ekman pumping ….
Comment: In McCreary et al., 1996 paper, at 8N, Equatorial forcing and Ekman pumping both look important for currents below 200m, though the current magnitude is very weak in both cases. The authors only say that ekman pumping is important without a convincing argument.

(2) We agree with the Reviewer, however there are presently no observations available that might support the argument, and so determining the mechanisms driving the undercurrent is beyond the scope of this study.

(3) *The discussion on possible driving mechanisms has been revised to clarify the point (P7, L7-10).*
*"The circulation in the southern BoB is complex as it is controlled by various forcings, such as local winds, Ekman pumping, and equatorial waves, which impact the region in different seasons. The mechanisms driving the undercurrent remain unclear and although beyond the scope of this study we put forward some informed hypotheses based on previous studies and our observations. The Rossby waves, equatorial waves, and Ekman pumping are likely to be important for producing the undercurrent."*

6. Page 6 Lines 30-35: The last paragraph of the discussion section, deals with currents in Pacific Ocean.
Comment: This seems completely out of context.

(2) This was also pointed out by Reviewer (1) in Reviewer Comment (10). Our original thought was to compare and contrast the Sri Lanka undercurrent with other low latitude boundary currents, such as the Mindanao Current. However, we agree with the reviewer that a comparison with undercurrents observed in the BoB is more relevant. In the revised text the undercurrent off the Sri Lankan east coast is now discussed in the context of the undercurrents along the Indian east coast (the EICC- reported in Mukherjee et al., 2014) and the Sri Lankan south coast (SMC and WMC- reported in Schott et al., 1994).